# HGF and VEGF-A and Their Receptors Show Expression and Angiogenic Effects on Human Choroidal Endothelial Cells: Implications for Treatments of Neovascular Age-Related Macular Degeneration

**Elizabeth A. Stewart, Claire L. Allen, Govindi J. Samaranayake, Thomas Stubington, Rukhsar Akhtar, Matthew J. Branch and Winfried M. Amoaku \***

Academic Ophthalmology, Division of Clinical Neuroscience, University of Nottingham, B Floor, Eye and ENT Building, Queen's Medical Centre, Nottingham NG7 2UH, UK; E.Stewart@nottingham.ac.uk (E.A.S.); claire.allen@exonate.com (C.L.A.); gjayanikasa@gmail.com (G.J.S.); mzydtjs@nottingham.ac.uk (T.S.); rukhsar.a@live.co.uk (R.A.); Matthew.Branch@nottingham.ac.uk (M.J.B.)
\* Correspondence: wma@nottingham.ac.uk; Tel.: +44-115-9249924 (ext. 64744)

**Abstract:** Intraocular neovascularisation is associated with common blinding conditions including neovascular age-related macular degeneration (nAMD). Vascular endothelial growth factor (VEGF) is central in driving choroidal neovascularisation in this disease. Many clinical therapies target VEGF-A with intravitreal anti-VEGF drugs, which, however, have limited efficacy and require repeated, prolonged treatment. Other cytokines are known to be involved, including hepatocyte growth factor (HGF), which is shown to have a role in the early stages of nAMD. We investigated the effect of HGF and its co-operation with VEGF-A on human choroidal endothelial cells (CEC). The expression of HGF and related molecules in CEC was investigated using immunofluorescence, Western blotting and flow cytometry. In vitro assays for proliferation, tubule formation and migration were used to assess the potential role of HGF in neovascularisation. Primary human CEC expressed HGF, VEGF-A and their receptors MET and VEGF receptor 2 (VEGFR2). HGF increased CEC proliferation, tubule formation and migration; the increased proliferation and migration appeared to be additive with that achieved with VEGF-A. This study provides insight into growth factor co-operation in CEC signalling and indicates that simultaneous blockage of multiple growth factors or common downstream signalling pathways may provide a more sustained treatment response, enhancing treatments in nAMD.

**Keywords:** hepatocyte growth factor (HGF); vascular endothelial growth factor (VEGF); age-related macular degeneration; neovascularisation; choroid; cytokine co-operation

## 1. Introduction

Age-related macular degeneration (AMD) is the most common cause of visual loss in the Western world, and its prevalence has been projected to increase significantly (15%) by 2050 [1,2]. The neovascular (wet) form of AMD (nAMD) due to choroidal neovascularisation (CNV) accounts for most of the severe visual loss associated with the disease. CNV is a form of angiogenesis where new blood vessels develop from the existing choriocapillaris. It involves endothelial cell (EC) receptor stimulation by cytokines, EC basement membrane proteolysis, dissociation of pericytes and smooth muscle cells and EC proliferation and migration [3]. Vascular endothelial growth factor (VEGF) has been identified as the most important cytokine driving CNV in nAMD [4,5]. As such, contemporary clinical management practice of nAMD involves the frequent administration of anti-VEGF drugs intravitreally. It is accepted that angiogenesis is a tightly regulated process that depends on a critical balance between multiple pro- and anti-angiogenic molecules [6]. In addition to VEGF, there are several other angiogenic growth factors present in the choroid and retinal pigment

epithelium (RPE) and associated with CNV [4]. A report from Cabral et al. [7] indicated that there was significant upregulation of other cytokines, including HGF following VEGF-A blockage with intravitreal injections of bevacizumab in nAMD eyes. Zhang et al. [8] reported that blockage of HGF and VEGF by a multi-kinase inhibitor of MET and VEGFR2 in a mouse model of CNV, through inhibition. Similarly, Roundabout 4 (ROBO4) (an inhibitor of angiogenesis and permeability) mRNA, was detected in choroidal vessels of nAMD patients, and an anti-ROBO4 antibody was reported to suppress migration of human umbilical vein endothelial cells (HUVEC) and CNV formation in a non-human primate model [9].

HGF stimulates the proliferation and migration of several cell types including EC [10–15]. In particular, Cai et al. [16] have shown that HGF and its receptor MET are expressed by bovine retinal endothelial cells (REC), with these cells exhibiting mitogenesis and migration upon HGF stimulation. HGF and MET are also highly expressed by human RPE cells and thought to be involved in neovascularisation of the retina and iris in diabetic eye disease [17–22]. However, information on its role in human choroidal disease is limited. Our laboratory discovered that the CD44 cell surface protein family, which has been shown to act as a co-receptor for MET and VEGFR2, was more highly expressed in human CEC than human retinal EC [23]. Du et al. [24] suggest that a potential inhibition of CNV may be achieved through decorin inhibition of angiogenesis via downregulating hypoxia-induced MET, Rac1, HIF-1$\alpha$ and VEGF expression in RPE cells. As such, blockade of other growth factors involved in CNV development and progression, in addition to anti-VEGF-A therapy, may result in more prolonged disease remission. It is also known that there is significant heterogeneity in gene expression profiles of EC derived from different species and tissue sites [23,25]. We have previously described the effects of other cytokines on CEC proliferation and in vitro angiogenesis [26].

There is, however, a paucity of information on HGF expression in the choroid and its relative expression in the human retina, choroid and RPE. Furthermore, no reports exist on the effects of HGF in vitro in human CEC. This paper primarily describes our investigation of the expression of HGF and MET in the choroid and compares CEC secretion of HGF and VEGF-A with that from REC and RPE cells and the effects of HGF and VEGF-A, alone or in combination, on CEC proliferation and angiogenesis in vitro.

## 2. Materials and Methods

Patients and the public were not involved in the design or conduct of the research. Research was undertaken with the approval of the Nottingham local research ethics committee (Nottingham 1, Q1060301, ver 6 dated 5 May 2011). Experiments were conducted in accordance with the tenets of the Declaration of Helsinki (Version 7, 2013) and complied with institutional regulations of good laboratory practice (GLP) and Health and Safety guidelines of the University of Nottingham. Aseptic conditions were followed in all experiments, which were performed within the laminar flow air hood (Envair, UK).

### 2.1. Cell Isolation and Culture

Human posterior segments free of any known ocular disease were obtained from the Manchester Eye Bank, UK. CEC and REC were isolated as previously described [27]. Donors were aged 26–69 years old and were non-diabetic. In brief, choroidal and retinal tissue was removed separately by dissection, treated with 0.1% (*w/v*) collagenase (Sigma-Aldrich, Poole, UK) in endothelial basal medium 2 (SFM) (Lonza, Wokingham, UK) for 1 h at 37 °C. For RPE cell culture, the RPE layer was brushed off the inner surface of Bruch's membrane then washed with sterile phosphate buffered saline (PBS) and cultured in DMEM:F12 with GlutaMax, with 25% foetal calf serum (FCS) and 1× penicillin amphotericin. EC were isolated using anti-CD31 coated Dynabeads (Life Technologies, Paisley, UK). Isolated EC were cultured on tissue culture plastic coated with 5 µg/mL fibronectin (Sigma) in EGM2-MV with hydrocortisone omitted (Lonza), in a humidified atmosphere of 5% $CO_2$, at 37 °C. EC purity was confirmed as previously described (expres-

sion of CD31/PECAM) [28]. Similarly, purity of RPE cells was confirmed by the expression of RPE cell markers (RPE65) and absence of fibroblast markers.

### 2.2. Immunohistochemistry

Whole choroidal tissue was removed by dissection, embedded in optimal cutting temperature (OCT) freezing compound (Leica, Germany) and frozen using liquid nitrogen. Seven-micrometre sections were prepared on 3-aminopropyltriethoxysilane coated slides and fixed with 100% acetone. Sections were stained with primary antibodies to MET (Abcam, Cambridge, UK, ab47431), VEGFR2 (ab39638), overnight, at 4 °C. Primary antibodies were visualised using the appropriate secondary antibody conjugated to fluorescein isothiocyanate (FITC) at 1:400, for 1 h at room temperature.

### 2.3. Immunocytochemistry

Isolated CEC were cultured on 8-well glass chamber slides (NUNC, UK) coated with 5 µg/mL fibronectin (Sigma) before fixing with 0.4% ($w/v$) paraformaldehyde and stained as described in Section 2.2; secondary antibodies were conjugated with Texas red. Staining of CEC was undertaken without permeabilisation.

### 2.4. Western Blotting

The expression of proteins of interest in cultured CEC were determined by Western blotting as previously denoted [27]. Primary CEC were lysed in 1xLDS sample buffer and separated on a 12% NuPAGE Bis tris 1.0 mm 12-well gel (Life Technologies) before transfer onto a polyvinylidene difluoride membrane (PVDF, Millipore imobilon-P Membrane, 0.45 µm). Membranes were probed using primary antibodies targeted to HGF (ab24865), VEGF-A (ab46154), MET (ab47431) and VEGFR2 (ab39638), followed by the appropriate alkaline phosphatase conjugated secondary antibody. Western blots were visualised using BCIP/NBT solution (Sigma).

### 2.5. Flow Cytometry

CEC were detached using 1xTrypLE (Life Technologies) for 5 min then transferred to flow cytometry tubes fixed in 3% ($v/v$) formaldehyde for 5 min at room temperature. This was followed by incubation with the appropriate primary antibody to MET or VEGFR2, at 1:50 in 1% ($w/v$) bovine serum albumin in PBS, overnight at 4 °C with agitation. Cells in each tube were washed then incubated with the appropriate FITC conjugated secondary antibody at 1:200 in 1% ($w/v$) BSA in PBS, for each secondary a corresponding control with cells was incubated in a separate tube. A tube of cells was identically prepared but without antibodies for the purposes of gating. After 30 min, cells were washed, centrifuged at $200 \times g$ for 10 min and re-suspended in 500 µL of PBS before analysis. Samples were analysed using the Epics Altra Flow Cytometer (Beckman Coulter, London, UK); 50,000 events were collected and analysed for each sample on the flow cytometer. Data obtained were further analysed on a dot plot and histogram using WEASEL software (www.wehi.edu.au; version 3.0, accessed on 8 April 2013), with cellular debris being excluded from the analysis through relevant gating of events according to the forward and side scatter. Secondary controls were used as negative controls to set the threshold (0.5%) for the percentage of positive cells.

### 2.6. Enzyme-Linked Immunosorbent Assay (ELISA)

Confluent CEC, REC and RPE cells were incubated in serum free medium (SFM) for 72 h, and the medium collected and sterilised through a 0.21 µm filter before ELISA. VEGF-A and HGF Quantikine® ELISA (R&D Systems) were performed as per the manufactures instructions.

## 2.7. Proliferation Assay

Proliferation of CEC was assessed using the WST-1 assay (Roche, Sussex, UK) as described and validated previously [26,27]. In brief, 2000 unpassaged CEC in EGM2-MV were added to the inside 60 wells of a fibronectin coated 96-well plate (BD Biosciences, Oxford, UK) and incubated for 24 h to allow cell attachment before washing with SFM, supplemented with antibiotics and 0.5% (*v/v*) 4 h heat inactivated serum, here after described as serum reduced medium (SRM) and incubated for 24 h. The medium in the first 6 wells was then replaced with SRM containing 1000 pM VEGF 165 or HGF (R&D Systems, Abingdon, UK) and serially diluted. Following incubation for 48 h, medium was replaced with 100 μL Cell-8 reagent (Sigma), incubated for 4 h at 37 °C, and the absorbance of each well was recorded at 650 nm using a Thermamax microplate reader (Molecular Devices, Wokingham, UK) according to the manufacturer's instructions. Cell proliferation was expressed as a percentage increase relative to SRM controls, examined in triplicate and repeated using CEC from at least 3 different donors.

## 2.8. Migration (Wound Closure) Assay

Confluent CEC on 6-well plates were serum starved overnight then a single scratch performed using a 200 μL pipette tip. Medium was changed to SFM or 500 pM HGF, 500 pM VEGF 165 or 500 pM each growth factor. These concentrations were determined based on preliminary optimisation studies. Images were taken at a marked point at time 0 then 24 h after the scratch. Images were taken and analysed using ImageJ software (http://imagej.nih.gov/ij/ accessed on 8 April 2013); % wound closure was calculated from the difference in scratch/wound area at 24 h versus 0 h.

## 2.9. In Vitro Tubule Formation

A 1:1 mixture of chilled Matrigel (BD Biosciences) and SFM, 2000 pM HGF, 2000 pM VEGF 165 or 2000 pM of each growth factor was dispensed into pre-chilled wells of an 8-well chamber slide. The Matrigel was allowed to solidify at 37 °C for 30 min before CEC suspended in SFM, 1000 pM HGF, 1000 pM VEGF 165 or 1000 pM of each growth factor (combined VEGF 165 and HGF) were seeded at a density of $4.8 \times 10^4$ per well. The wells were observed hourly for the formation of tubes, which were then imaged and counted for the formation of tubules and branch points (BP). Tubules were also stained with 10 μM calcein AM (BD Biosciences) and 300 nM DAPI for 10 min each for visualisation on a fluorescence microscope (Olympus BX51) and imaged using Cell^F software (Olympus, UK).

## 2.10. Statistical Analysis

All experiments were repeated with at least 3 donors. An unpaired t-test was used to compare data from multiple donors. For all analyses, *p* values below 0.05 were considered indicative of significant difference.

## 3. Results

### 3.1. Expression of VEGFR2 and MET in Human Choroid and Choroidal Endothelial Cells

In vivo expression of VEGFR2 (Figure 1B) and MET (Figure 1C) by immunofluorescent staining of whole choroid sections were demonstrated in the choroid compared to the appropriate secondary antibody controls (Figure 1A,D). VEGFR2 was seen to be localised mainly to vessels, whereas MET staining was more diffuse throughout the choroid tissue. Expression of VEGFR2 (Figure 1E) and MET (Figure 1F) was also detected in cultured CEC. CEC were fixed but not permeabilised before staining so that the images show surface expression. Therefore, the immunofluorescent images for MET were slightly brighter as shown despite being visualised using the same settings.

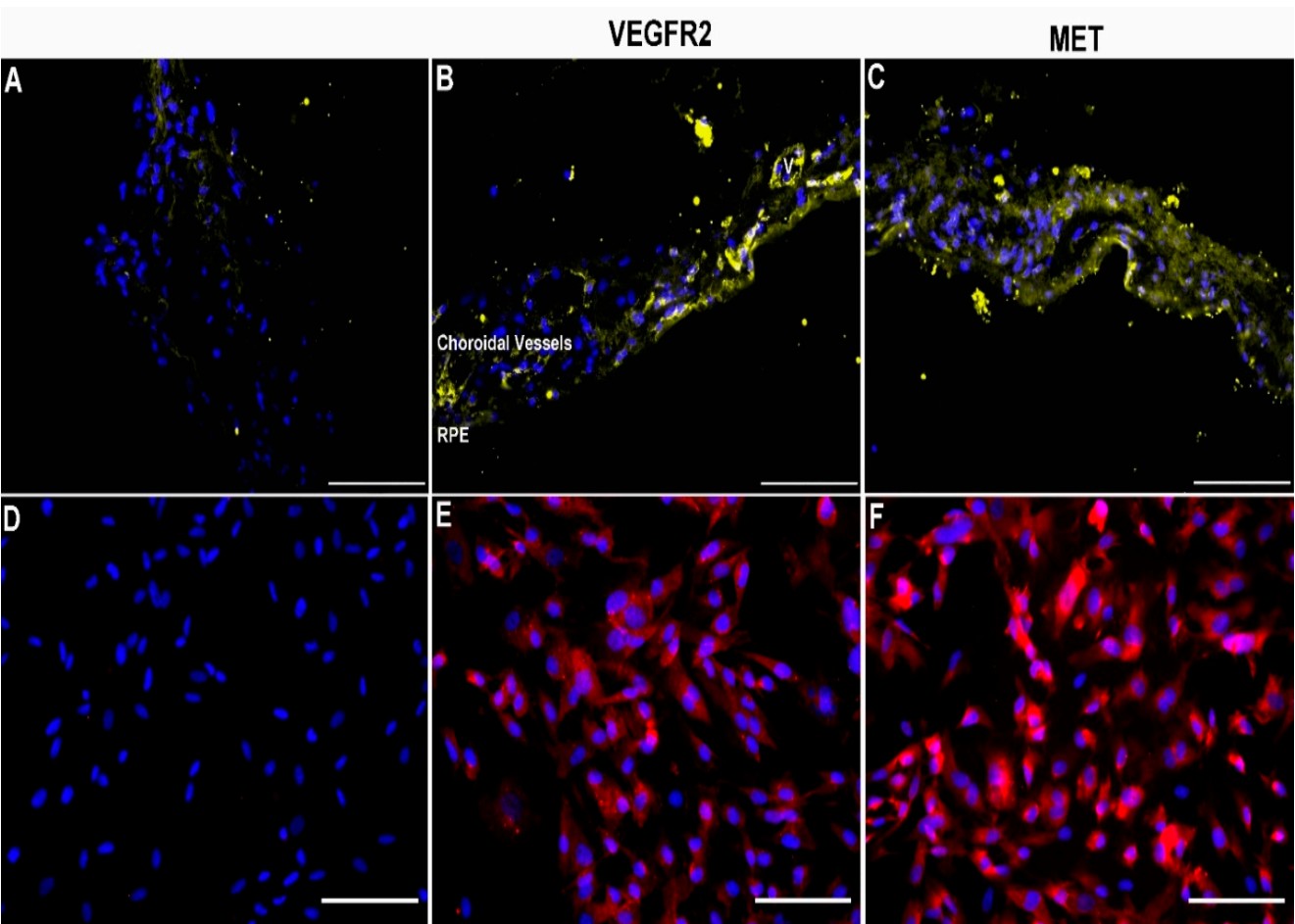

**Figure 1.** Immunofluorescence staining indicated possible higher expression of MET than VEGFR2. Sections of whole human choroid were stained with specific antibodies (**A–E**) to MET (**B,E**), VEGFR2 (**C,F**) a secondary control was also performed (**A,D**), vessels are marked 'v' and scale bar is 100 μm.

Flow cytometry quantification of gated live cells (Figure 2A) showed surface expression of the receptors of cultured CEC in normal growth culture conditions and showed that 79.9% ± 15.25 expressed detectable levels of VEGFR2 (Figure 2D,F) over control IgG (Figure 2B) and 73.1% ± 17.04 expressed MET (Figure 2C,F) along with the known co-receptor CD44v6 (Figure 2E,F).

### 3.2. CEC Secrete High Levels of VEGF and HGF

All three cell types (CEC, REC and RPE) produced and secreted endogenous growth factors (Figure 3) as detected by ELISA. Quantified levels of VEGF-A and HGF released into the media after 72 h are shown in Figure 3A,B. CEC and REC produced similar levels of VEGF-A (CEC: 0.49 ng/mL ± 0.29, REC: 0.68 ng/mL ± 0.52), whilst VEGF-A production by RPE cells was more variable, but higher on average (2.01 ± 1.72). Conversely, CEC produced significantly more HGF (18.56 ng/mL ± 2.47) than either REC (6.97 ng/mL ± 4.68, $p = 0.0121$) or RPE (9.91 ng/mL ± 3.67, $p = 0.0301$). The release of HGF by CEC was also found to be time-dependent and sustained; a significant ($p = 0.0015$) increase over 5 days was observed (Figure 3C).

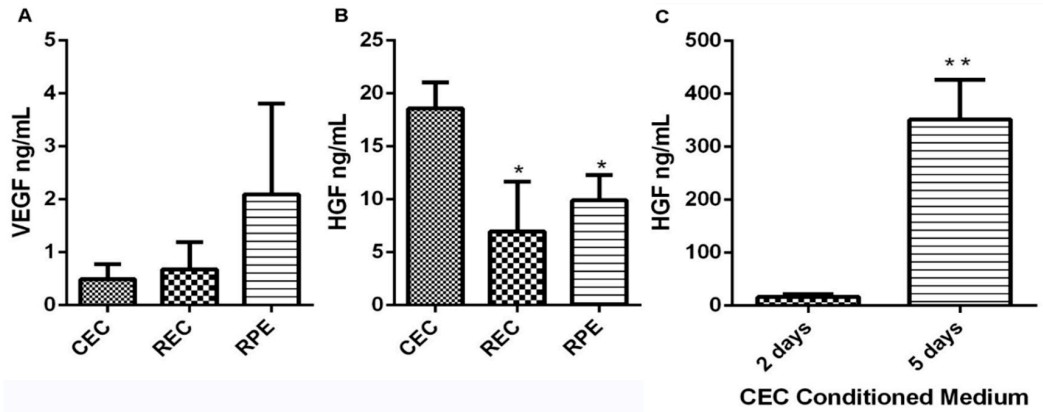

**Figure 2.** Expression and surface localisation of MET and VEGFR2 in cultured human CEC. CEC were fixed and stained with specific primary antibodies and a secondary FITC conjugate. Representative dot plots of flow cytometry gating (**A**) secondary control (**B**), surface expression of MET (**C**), VEGFR2 (**D**), CD44v6 (**E**); flow cytometry was performed on at least 3 donors % expression plotted (**F**) mean ± SD.

**Figure 3.** Secretion of VEGF-A and HGF by primary human CEC, REC and RPE. ELISA was used to quantify VEGF (**A**) and HGF (**B**) secreted over 72 h. HGF was also quantified after 2 and 5 days (**C**). Graphs show mean ±SD, *n* = 3; * *p* < 0.05, ** *p* < 0.005).

### 3.3. HGF Stimulation of CEC Proliferation and Migration

HGF stimulated increased proliferation of CEC with a similar potency to VEGF at concentrations as low as 7.8 pM. An additive effect was seen on CEC proliferation, when VEGF-A and HGF were applied in combination; this change was statistically significant at 7.8 pM ($p$ = 0.0306) (Figure 4). When a wound healing assay was performed (Figure 5), wound closure was significantly increased by HGF ($p$ = 0.0319), VEGF-A ($p$ = 0.0041) and HGF + VEGF-A ($p$ = 0.0094), compared to unstimulated. Again, the growth factors appeared to be more potent in promoting wound closure when used in combination, but this effect was not significantly greater than either of the growth factors alone.

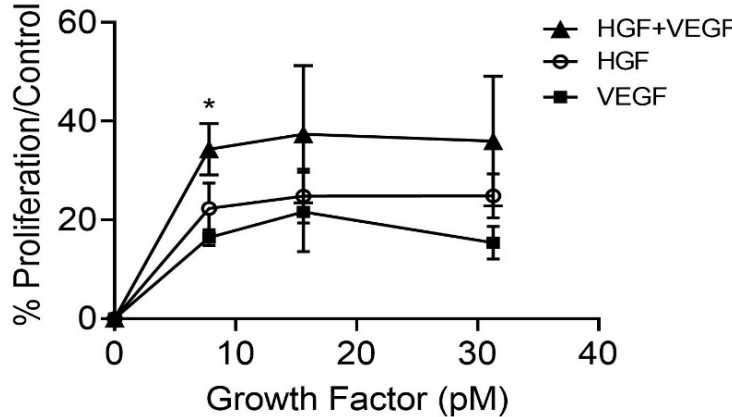

**Figure 4.** HGF stimulated CEC proliferation with a similar potency to VEGF-A. CEC from at least 3 donors were exposed in triplicate to HGF, VEGF-A, or a combination. Both HGF and VEGF-A stimulated CEC proliferation with similar efficiencies, and a general additive effect was seen when the growth factors were combined (* $p$ = 0.0306).

### 3.4. Effect of HGF on CEC Tubule Formation on Matrigel

Tubule formation and BP were observed with the CEC in Matrigel and SFM. The addition of HGF significantly increased the number of BP ($p$ = 0.0461) and tubules ($p$ = 0.0014) compared to Matrigel alone (Figure 6). Similarly, VEGF 165 increased BP (ns, $p$ = 0.1091) and significantly increased tubule formation ($p$ = 0.0364). When treated with both VEGF 165 and HGF, tubules were increased over SFM alone over some of the wells; however, areas of cell monolayer were visible (Figure 6, arrow). Optimisation was attempted by reduction in seeding density in the well and reduction in the concentration of each growth factor to 500 pM and then 250 pM of each, but this did not prevent these patches of non-tube and proliferating CEC from forming.

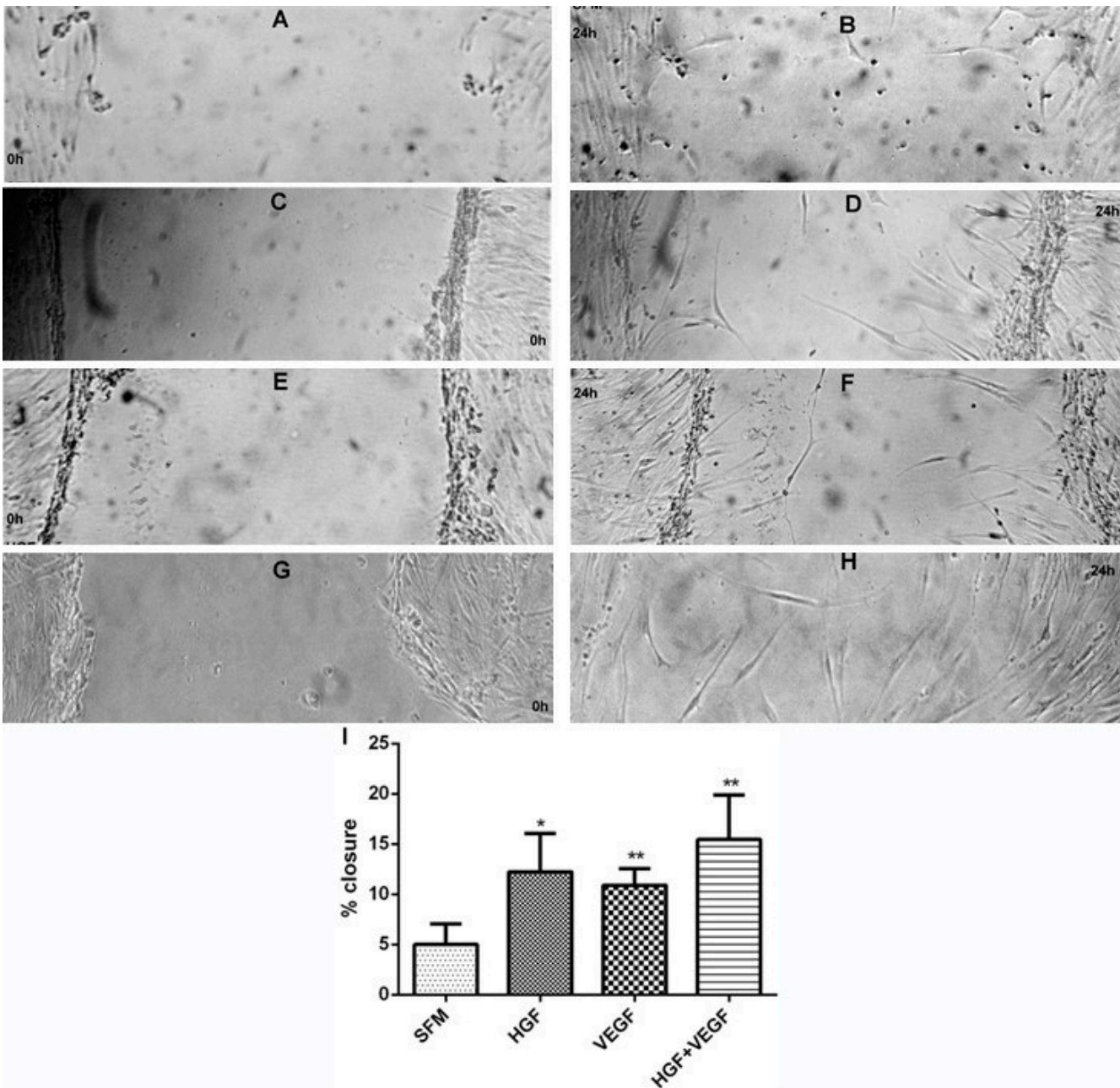

**Figure 5.** HGF stimulated migration (wound closure) was equivalent to VEGF-A. Confluent CEC were injured then incubated in EBM for 24 h (**A**,**B**) or 500 pM VEGF-A (**C**,**D**), 500 pM HGF (**E**,**F**) or HGF and VEGF-A combination (**G**,**H**). Scratch closure was imaged and quantified after 24 h, plots (**I**) represent % closure, mean ± SD ($n = 4$, * $p < 0.05$, ** $p < 0.005$).

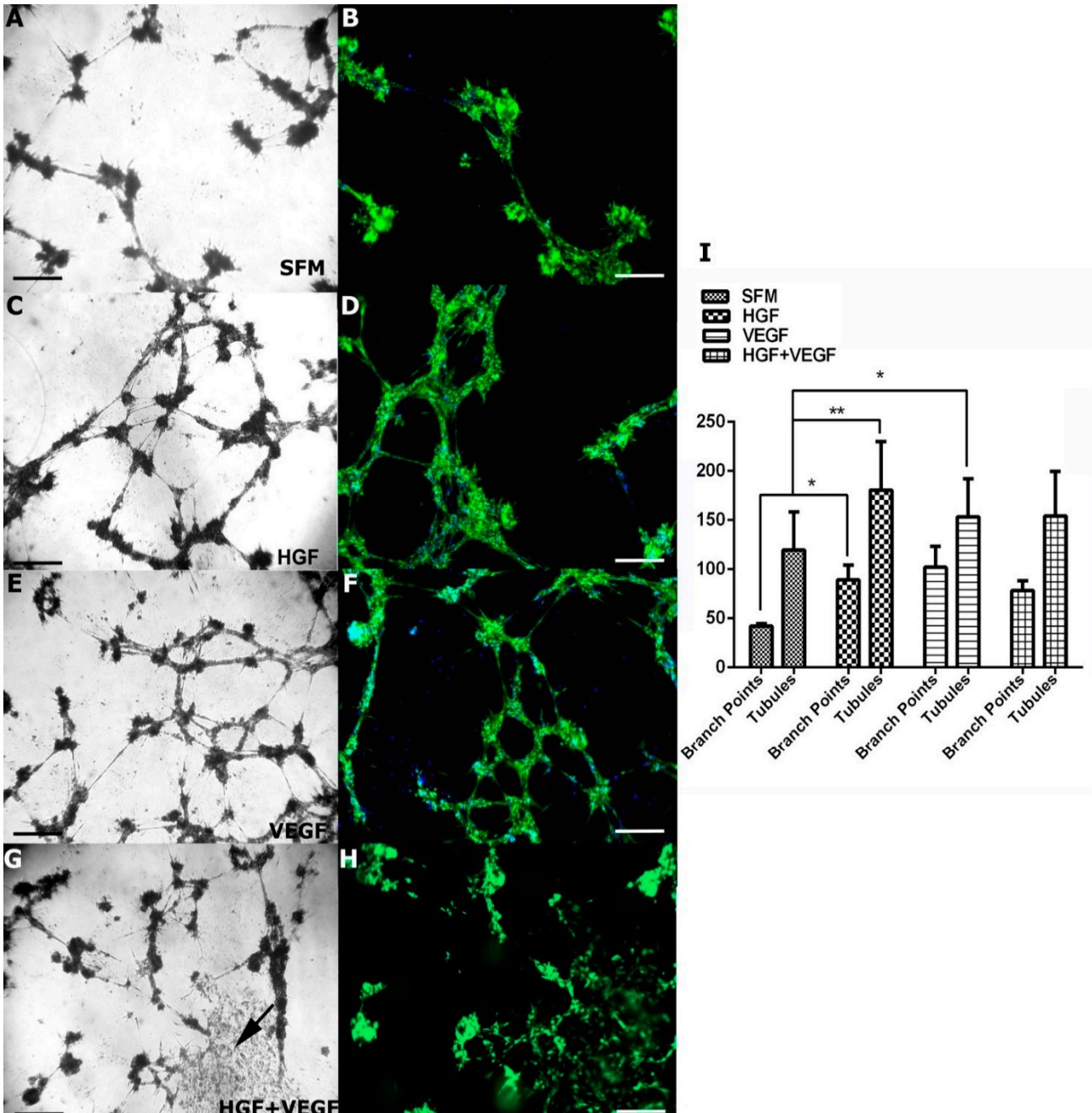

**Figure 6.** HGF stimulated CEC tubule formation and branching with similar efficiency to VEGF 165. CEC were seeded in EBM (**A,B**) or EBM containing 1000 pM HGF (**C,D**) or VEGF 165 (**E,F**) or HGF and VEGF combined (**G,H**) on Matrigel containing the same concentration of GF Cells were observed hourly until tubules were formed, and imaging using phase contrast (**A,C,E,G**), and calcien AM and DAPI staining (**B,D,F,H**) arrows indicate areas of proliferation. Number of branch points and tubules counted and plotted (**I**). Scale bar is 25 M. (* $p < 0.05$; ** $p < 0.036$).

## 4. Discussion

Apart from VEGF, other cytokines including HGF, bFGF and platelet-derived growth factor (PDGF) have also been shown to play key roles in the development and progression of CNV in many laser-induced CNV animal models [29,30]. MET expression and HGF effects have mainly been studied in tumours [31–34] and experimental cell lines [35]. In the posterior segment disease of the eye, HGF has previously been shown to be upregulated in laser-induced CNV in animal models [30] and present in the aqueous humour of eyes

with nAMD [36]. RPE cells have also been shown to be responsive to HGF [20], through expression of MET, the only known receptor of HGF [37]. Similarly, HGF has been evaluated in eyes with diabetic eye disease [38,39]. However, the role of HGF in choroidal vascular disease—particularly nAMD—has not been studied in detail, despite the detection of HGF in the early stages of CNV [30] and reports of CNV suppression in mouse and non-primate models [9,40]. To the best of our knowledge, this present study is the first to detect the expression of MET in sections of the human choroid, as well as primary CEC cultures, and in comparison to REC and RPE. MET was expressed on the surface of CEC, indicating a functional presence on these cells. In addition, we have detected endogenous HGF in CEC, indicating these cells have the ability to express HGF (in addition to VEGF), at a much higher level than found in REC and RPE cells. The higher levels of endogenous HGF in CEC may partly explain fenestrations in the choriocapillaris. This finding further suggests that in pathological states, exogenous HGF, produced by the RPE [20], in addition to VEGF [41], as well as endogenous HGF (from CEC) may activate MET in CEC. Similarly, a reduction in HGF production by CEC and RPE may result in the demise of CEC, as occurs in dry AMD. Furthermore, HGF, amongst other cytokines, have been reported as upregulated after VEGF blockage with intravitreal injections of bevacizumab in eyes with nAMD [7].

HGF has been reported to be a potent mitogen and increases proliferation in several cell types including EC [10,16,21]. In this study, HGF was found to be as potent at stimulating proliferation of CEC as VEGF isoform 165. Donor variability prevented an obvious dose response on proliferation. As previously reported by our group, FGF2 and VEGF [26] in combination resulted in a greater stimulation of proliferation, indicating separate or additive signalling mechanisms with such combination. This additive effect on proliferation, although not significant, was also observed whilst studying angiogenesis with HGF and VEGF 165. Despite previous assumptions that VEGF 165 is the most potent angiogenic factor for these cells [42–44], in primary CEC cultures, HGF is as potent in stimulating proliferation and angiogenesis. The potential additive effect of HGF and VEGF-A on CEC proliferation indicates that HGF can induce proliferation independently of VEGF-A [45] irrespective of any downstream of potential downstream signalling of HGF through VEGF [31,46,47]. HGF or VEGF-A in isolation significantly increased tubule formation and branching by CEC. However, when HGF was combined with VEGF, areas of cell monolayers appeared, indicating that tubule formation and branching were not significantly enhanced by combination of both cytokines compared to that achieved by each on its own. This effect was not altered by seeding density or growth factor concentration and suggested that the combination of VEGF 165 and HGF appeared to preferentially stimulate proliferation over microtubule formation. The combination of HGF and VEGF 165 did appear to have a slight additive effect on EC migration; however, this effect was not significant. The implications are that there are at least two separate mechanisms of HGF action and that proliferation and migration have separate mechanisms [45–47].

Current anti-VEGF therapies have the potential to remove functional as well as pathological levels of VEGF from the choroid and retina for 4–8 weeks (depending on the particular anti-VEGF drug). These treatments therefore require frequent repetition. Furthermore, it is now recognised that not all CNV regress with anti-VEGF blockage. In addition, some cases become less responsive after demonstrating good initial response. Concerns have also been expressed over the long-term effects of VEGF starvation on endothelial cells and the neuroretina [48,49]. This may increase underlying ischaemia and lead to compensation by other growth factors [24,50,51].

Research in oncology has determined that HGF can be as potent a mitogenic and angiogenic factor as VEGF-A [47]. HGF is also known to inhibit thrombospondin 1, a well-described angiostatic molecule [46]. As such, blockage of HGF may have the dual role of reducing angiogenic stimulation as well as promoting angiostatic actions of thrombospondin 1. Despite the initial responsiveness of different tumours to anti-VEGF therapies, tumour growth resumes despite anti-VEGF inhibition because alternative

pathways including HGF-MET signalling are quickly adopted by tumour cells [51–54]. As well as VEGF-A antagonism, HGF blockade with peptide derivatives of HGF, for example, NK4 (pre-clinical phase) [32,55] and H-RN [56], and HGF-neutralising antibodies such as rilotumumab (phase II clinical trial) [57,58] has been studied in angiogenesis reduction. Similarly, as in oncogenesis, it is suggested that blockade of other growth factors involved in CNV development and progression, in addition to anti-VEGF therapy, may result in more prolonged disease remission.

This study suggests that cytokines other than VEGF-A, including HGF, are important in the development of CNV and its control, and that crosstalk between HGF and VEGF-A (and other cytokines, such as FGF) may occur. This is further supported by the recent report by Cabral et al. [7] and the mouse and nonhuman primate experimental CNV models [9,40]. As such, adoption of combination therapies that block both VEGF-A and HGF may result in more permanent blockage of CNV growth, as has been suggested in cancer therapy [51,53,59]. A more efficient approach to regulate intraocular angiogenesis would aim to simultaneously block the actions of multiple growth factors or, if feasible, the common downstream site in the angiogenesis pathway.

A limitation of this study is that we did not investigate whether HGF signalling through MET affects autocrine secretion of VEGF or CEC response to MET and/or VEGFR2 knockdown. Inhibition of HGF or VEGF would validate the biological data further and is planned for future experimentation. Further studies investigating the specific targeting of co-operative signalling in CEC may lead to more effective prevention and treatment of angiogenesis and ultimately improved therapies for CNV. The differential response of CEC, REC or RPE to HGF treatment is beyond the scope of the current study, but planned.

## 5. Conclusions

HGF, VEGF-A and their receptors MET and VEGFR2 are expressed in primary human CEC. HGF and VEGF-A had additive effects on CEC proliferation and migration. Similarly, tube formation was enhanced with the two growth factors. Co-operation between HGF and VEGF-A occurs in CEC signalling and indicates that simultaneously blocking the actions of multiple growth factors, or the common downstream signalling pathway, may provide a more sustained treatment response in order to enhance treatment protocols in nAMD.

**Author Contributions:** Conceptualization, E.A.S. and W.M.A.; experimentation, E.A.S., G.J.S., C.L.A., M.J.B., T.S. and R.A.; data analysis, all authors; writing—original draft preparation, E.A.S. and W.M.A.; writing—review and editing, E.A.S. and W.M.A.; funding acquisition, W.M.A. All authors have read and agreed to the published version of the manuscript.

**Funding:** This research was funded by an unrestricted research grant from Novartis Pharma, UK.

**Institutional Review Board Statement:** Patients and the public were not involved in the design or conduct of the research. Research was undertaken with the approval of the Nottingham local research ethics committee (Nottingham 1, Q1060301, ver 6 dated 5 May 2011). Experiments were con-ducted in accordance with the tenets of the Declaration of Helsinki (Version 7, 2013) and complied with institutional regulations of good laboratory practice (GLP) and Health and Safety guidelines of the University of Nottingham. Aseptic conditions were followed in all experiments, which were performed within the laminar flow air hood (Envair, UK).

**Informed Consent Statement:** Not applicable.

**Conflicts of Interest:** W.M.A.: Commercial relationships: acted as consultant for Abbvie, Alimera, Allergan Inc., Bayer, Apellis, Novartis, Pfizer, Santen and Thrombogenics and has undertaken research sponsored by Allergan, Bayer, Boerhinger Ingelheim, Novartis and Pfizer. He has received speaker fees and travel grants from Allergan, Bausch and Lomb, Bayer, Novartis and Pfizer. All other authors declare no conflict of interest. The funders had no role in the design of the study; in the collection, analyses, or interpretation of data; in the writing of the manuscript, or in the decision to publish the results.

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
