# Peer review of "HGF and VEGF-A and Their Receptors Show Expression and Angiogenic Effects on Human Choroidal Endothelial Cells: Implications for Treatments of Neovascular Age-Related Macular Degeneration"

_2673-8937, doi:10.3390/ijtm1010006_

Round 1

Reviewer 1 Report

In this work, Stewart and colleagues analyze the effects of HGF and VEGF (which one? VEGF-A?) on human choroidal endothelial cells. The angiogenic role of the VEGF family of growth factors is already known, and this work does not add new information.

At the end of the first paragraph of the Introduction, authors cite a paper (three-years old, it cannot be considered “recent”) in which upregulation of HGF, among other cytokines, was observed in the aqueous humour of patients after administration with an anti-VEGF-A therapy. Authors should better explain why they have chosen to further investigate only HGF.

I would find more interesting a deeper evaluation of cytokines and receptor expression. Retinal (REC) and choroidal (CEC) endothelial cells express also VEGFR-1, another receptor for VEGF-A and a receptor for VEGF-B and placenta growth factor (PlGF). Could the author also investigate this expression in the endothelial cells they used here in vitro and in vivo? Why neither REC nor retinal pigment epithelial cells (RPE) were analysed by FACS in Figure 2?

Could either treatment of CEC with HGF increase secretion of VEGF-A or vice versa, treatment with VEGF-A further increase expression and secretion of HGF? Do VEGF-A or HGF treatments enhance receptor expression? Does bevacizumab inhibit HGF or MET expression or any of these HGF-mediated biological properties in CEC? Inhibition of HGF should also be performed to validate the biological data. Do CEC, REC or RPE respond differently to HGF treatment? I understand that the authors would perform additional experiments in the future on this topic, but some of these questions should be answered in this paper.

Last year, two different articles proposed the usage of therapeutic molecules that target different cytokine signalling, MET and VEGF  receptors as well, for treatment of choroidal neovascularization. Since a possible combined therapy is proposed by the authors in the discussion, these two articles should be cited and commented (Zhang X et al. Journal of Ophtalmology 2020, article ID 5905269; Isumi Y et al. Trans Vis Sci Tech 2020, 9:7).

Author Response

We thank the reviewers very much for their comments.

We have addressed these comments in the revised manuscript, and hope these satisfactorily address all the relevant points raised. These are addressed point by point below, and incorporated into the manuscript.

Reviewer 1

In this work, Stewart and colleagues analyze the effects of HGF and VEGF (which one? VEGF-A?) on human choroidal endothelial cells. The angiogenic role of the VEGF family of growth factors is already known, and this work does not add new information.

We agree that the effects of VEGF family of growth factors are well known (Frank et al, 1996; Kvanta et al, 1996). That is underscored by our earlier reports on the effects of different cytokines on human retinal (hREC) and choroidal endothelial cells (hCEC) (Browning et al, 2005, 2008) and others. We are also aware of the 2 recent publications on HGF referred to by the reviewer, and have incorporated them into the Introduction as well as Discussion. However, there is paucity of information on the expression of HGF by hREC, hCEC and RPE. Essentially, what our current manuscript adds is information on the relative expression of HGF and its receptor MET by these key cells in the ocular posterior segment, and effects of HGF and the combination of HGF and VEGF 165 on hCEC, and their potential role in disease.

At the end of the first paragraph of the Introduction, authors cite a paper (three-years old, it cannot be considered “recent”) in which upregulation of HGF, among other cytokines, was observed in the aqueous humour of patients after administration with an anti-VEGF-A therapy. Authors should better explain why they have chosen to further investigate only HGF.

We appreciate the reviewers comment on the word ‘recent’, which we have omitted. As explained above, investigation of the effects of HGF in combination with VEGF on hCEC is warranted in order to achieve better treatments for posterior segment disease.

I would find more interesting a deeper evaluation of cytokines and receptor expression. Retinal (REC) and choroidal (CEC) endothelial cells express also VEGFR-1, another receptor for VEGF-A and a receptor for VEGF-B and placenta growth factor (PlGF). Could the author also investigate this expression in the endothelial cells they used here in vitro and in vivo? Why neither REC nor retinal pigment epithelial cells (RPE) were analysed by FACS in Figure 2?

As explained above, our previous studies have described the effects of different cytokines other than VEGF on hREC and hCEC. This information has been included in the revised Introduction, in order to clarify the rationale for this study. Western blotting detected the total cell expression of VEGFR2, MET and ligands HGF and VEGF in primary hCEC. In addition, we have provided data on the secretion of VEGF and HGF by primary human CEC, REC and RPE in Fig 3, which we believe complements the FACS data shown in Fig 2.

Could either treatment of CEC with HGF increase secretion of VEGF-A or vice versa, treatment with VEGF-A further increase expression and secretion of HGF? Do VEGF-A or HGF treatments enhance receptor expression? Does bevacizumab inhibit HGF or MET expression or any of these HGF-mediated biological properties in CEC? Inhibition of HGF should also be performed to validate the biological data. Do CEC, REC or RPE respond differently to HGF treatment? I understand that the authors would perform additional experiments in the future on this topic, but some of these questions should be answered in this paper.

We have not attempted blocking HGF in our experiments. However, VEGF inhibition with ranibizumab/bevacizumab blocked the effects of VEGF in the combination treatment of hCEC in preliminary experiments. These will require further validation in the future, but our laboratory is unable to provide at the present because of Covid-19 issues.

We have not studied the effects of HGF on REC or RPE as we believed that is outside the rationale for the current study.

Last year, two different articles proposed the usage of therapeutic molecules that target different cytokine signalling, MET and VEGF receptors as well, for treatment of choroidal neovascularization. Since a possible combined therapy is proposed by the authors in the discussion, these two articles should be cited and commented (Zhang X et al. Journal of Ophtalmology 2020, article ID 5905269; Isumi Y et al. Trans Vis Sci Tech 2020, 9:7).

We thank the reviewer for the 2 references provided, and have appropriately incorporated them into the manuscript (Introduction and Discussion).

Reviewer 2 Report

Review of “HGF and VEGF and their receptors show expression and angiogenic effects on human choroidal endothelial cells: implications for treatments of neovascular age-related macular degeneration”

Comments:

  1. Abstract is very well-written, and shows a clear demonstration of the reasoning for experimentation, as well as the achieved results.
  2. Correct grammar in lines 44-45 “…dissociation of pericytes and smooth muscle cells and EC proliferation and migration.”
  3. Add either a comma or an m-dash to line 53 before the word “including”.
  4. Correct grammar on lines 55-56 “HGF and MET are also highly expressed by human RPE cells, and is thought to be…”
  5. Eliminate extra space in line 62 “and compares  CEC secretion”.
  6. There is no punctuation at the end of line 64.
  7. Add “the” before “public” in line 67.
  8. Change “were” to “was” to correct grammar in line 83.
  9. Change “embedding” to “embedded” in line 87.
  10. Fix grammar in lines 93-95 “Isolated CEC were…”
  11. Replace comma with semicolon in line 111.
  12. Add “of” before “each” in line 140.
  13. Add “of” before “each” in line 142.
  14. Results section is well-organized and easy to understand.
  15. Add comma before “HGF” in line 214.
  16. Add “and” before “platelet-derived” in line 229.
  17. Denote “particularly nAMD” with m-dashes instead of a comma and a space in line 235.
  18. Reword sentence on lines 242-243 “Similarly, reduced HGF production…”. End of sentence is currently difficult to understand with lack of comma before “as occurs in…”.
  19. Correct grammar on lines 243-245 “Furthermore, HGF, amongst other cytokines have…”
  20. Correct grammar on lines 248-249 “…FGF2 and VEGF, in combination resulted…”
  21. Add comma to line 252 after “cultures”.
  22. Add m-dash between “appeared” and “indicating” on line 256.
  23. Quantify “for a while” on line 264. How long is this?
  24. Correct grammar on line 269 “…as potent a mitogen…”.
  25. Add comma after “such” on line 270.
  26. Add comma after “HGF” on line 279.
  27. Remove comma after “HGF” on line 280.
  28. Add comma after “such” on line 281.
  29. Add comma after “growth” on line 282.
  30. Correct placement of commas on lines 282-284 “A more efficient approach…”.
  31. Correct spelling of “and” on line 286 “MET anmd/or VEGFR2”.
  32. Remove comma after “proliferation” on line 291.
  33. Denote “or the common downstream signalling pathway” on line 293 with m-dashes instead of comma.

Author Response

All the corrections suggested by Reviewer 2 below have been incorporated into the manuscript.

Comments:

  1. Abstract is very well-written, and shows a clear demonstration of the reasoning for experimentation, as well as the achieved results.
  2. Correct grammar in lines 44-45 “…dissociation of pericytes and smooth muscle cells and EC proliferation and migration.”
  3. Add either a comma or an m-dash to line 53 before the word “including”.
  4. Correct grammar on lines 55-56 “HGF and MET are also highly expressed by human RPE cells, and is thought to be…”
  5. Eliminate extra space in line 62 “and compares CEC secretion”.
  6. There is no punctuation at the end of line 64.
  7. Add “the” before “public” in line 67.
  8. Change “were” to “was” to correct grammar in line 83.
  9. Change “embedding” to “embedded” in line 87.
  10. Fix grammar in lines 93-95 “Isolated CEC were…”
  11. Replace comma with semicolon in line 111.
  12. Add “of” before “each” in line 140.
  13. Add “of” before “each” in line 142.
  14. Results section is well-organized and easy to understand.
  15. Add comma before “HGF” in line 214.
  16. Add “and” before “platelet-derived” in line 229.
  17. Denote “particularly nAMD” with m-dashes instead of a comma and a space in line 235.
  18. Reword sentence on lines 242-243 “Similarly, reduced HGF production…”. End of sentence is currently difficult to understand with lack of comma before “as occurs in…”.
  19. Correct grammar on lines 243-245 “Furthermore, HGF, amongst other cytokines have…”
  20. Correct grammar on lines 248-249 “…FGF2 and VEGF, in combination resulted…”
  21. Add comma to line 252 after “cultures”.
  22. Add m-dash between “appeared” and “indicating” on line 256.
  23. Quantify “for a while” on line 264. How long is this?
  24. Correct grammar on line 269 “…as potent a mitogen…”.
  25. Add comma after “such” on line 270.
  26. Add comma after “HGF” on line 279.
  27. Remove comma after “HGF” on line 280.
  28. Add comma after “such” on line 281.
  29. Add comma after “growth” on line 282.
  30. Correct placement of commas on lines 282-284 “A more efficient approach…”.
  31. Correct spelling of “and” on line 286 “MET anmd/or VEGFR2”.
  32. Remove comma after “proliferation” on line 291.
  33. Denote “or the common downstream signalling pathway” on line 293 with m-dashes instead of comma.

Round 2

Reviewer 1 Report

I appreciate the effort of the authors to answer to my questions and, in my opinion, the paper is more understandable now. Some spelling errors are still present.

However, additional experiments I requested were not provided, and I am afraid this work does not add any significant data in the field.

A minor point: Please, can you change VEGF into VEGF-A?